# Oral Immune-Related Adverse Events Caused by Immune Checkpoint Inhibitors: Salivary Gland Dysfunction and Mucosal Diseases

**DOI:** 10.3390/cancers14030792

**Published:** 2022-02-04

**Authors:** Yoshiaki Yura, Masakazu Hamada

**Affiliations:** Department of Oral and Maxillofacial Surgery, Osaka University Graduate School of Dentistry, Suita, Osaka 565-0871, Japan; hmdmskz@dent.osaka-u.ac.jp

**Keywords:** oral cavity, head and neck squamous cell carcinoma, immune checkpoint inhibitor, immune-related adverse event, cellular and humoral tumor immunity, Sicca syndrome, oral lichenoid reaction, pemphigoid

## Abstract

**Simple Summary:**

The antitumor effect of immune checkpoint inhibitors (ICIs) such as antibodies against CTLA-4, PD-1, and PD-L1 is higher than that of conventional chemotherapy and is durable, improving the survival of patients with advanced head and neck squamous cell carcinoma (HNSCC). However, this therapy reduces the immune tolerance state and allows antitumor cytotoxic CD8^+^ T cells to attack normal cells expressing self-antigens that cross-react with tumor antigens, and so it can induce immune-related adverse events (irAEs). Treatment of various malignancies, including HNSCC, with ICIs may result in the appearance of oral irAEs. In the oral cavity, irAEs such as Sicca syndrome, oral lichenoid reaction (OLR), and pemphigoid occur. It is necessary to elucidate the pathogenic mechanisms of these intractable diseases. Early diagnosis and appropriate approaches to irAE are needed for efficient treatment of advanced HNSCC by ICIs.

**Abstract:**

Conventional chemotherapy and targeted therapies have limited efficacy against advanced head and neck squamous cell carcinoma (HNSCC). The immune checkpoint inhibitors (ICIs) such as antibodies against CTLA-4, PD-1, and PD-L1 interrupt the co-inhibitory pathway of T cells and enhance the ability of CD8^+^ T cells to destroy tumors. Even in advanced HNSCC patients with recurrent diseases and distant metastasis, ICI therapy shows efficiency and become an effective alternative to conventional chemotherapy. However, as this therapy releases the immune tolerance state, cytotoxic CD8^+^ T cells can also attack organs and tissues expressing self-antigens that cross-react with tumor antigens and induce immune-related adverse events (irAEs). When patients with HNSCC are treated with ICIs, autoimmune diseases occur in multiple organs including the skin, digestive tract, endocrine system, liver, and respiratory tract. Treatment of various malignancies, including HNSCC, with ICIs may result in the appearance of oral irAEs. In the oral cavity, an oral lichenoid reaction (OLR) and pemphigoid develop. Sicca syndrome also occurs in association with ICIs, affecting the salivary glands to induce xerostomia. It is necessary to elucidate the pathogenic mechanisms of these intractable diseases that are not seen with conventional therapy. Early diagnosis and appropriate approaches to irAEs are needed for efficient treatment of advanced HNSCC by ICIs.

## 1. Introduction

A notable advance in the treatment of head and neck squamous cell carcinoma (HNSCC) in recent years has been the development of immunotherapy with immune checkpoint inhibitors (ICIs) using anti-CTLA-4, anti-PD-1, and anti-PD-L1 antibodies. Nivolumab and pembrolizumab are FDA-approved anti-PD-1 antibodies against HNSCC [1]. In resectable oral squamous cell carcinoma in HNSCC, radical resection and reconstructive surgery are usually performed, although radical cure by heavy ion radiotherapy has been also attempted [2,3,4,5]. If risk factors for recurrence are confirmed after surgery, additional chemoradiotherapy is given [6]. However, even with these treatments, recurrence occurs in 30–50% of cases [6,7,8]. For patients with recurrent and distant metastasis, cisplatin-based chemotherapy and/or the anti-EGFR antibody cetuximab is often used [9,10,11]. It is difficult to obtain a sufficient therapeutic effect in advanced stages by these conventional therapies. Moreover, re-irradiation for recurrent lesions is prone to causing serious dysfunction in eating and swallowing [12].

The activity of cytotoxic CD8^+^ T cells that recognize tumor antigens and attack tumor cells is regulated by co-stimulatory and co-inhibitory signals, in which CTLA-4-CD80/86 and PD-1-PD-L1 interactions form co-inhibitory pathways [13]. As anti-CTLA-4, anti-PD-1, and anti-PD-L1 antibodies interrupt these co-inhibitory signal pathways and restore the function of cytotoxic CD8^+^ T cells, these antibodies have been used to treat many chemo-resistant cancers. The ICIs improve the outcomes of patients with malignancies including melanoma, lung cancer, and head and neck cancer, and the range of indications is further expanding [14,15,16]. In HNSCC, the effects of ICIs have been investigated for patients with recurrent disease and/or distant metastasis, showing resistance to cisplatin-based chemotherapy. The treatment response was found to be higher than that of chemotherapy and durable, leading to improved patient survival [17,18,19,20,21,22]. Therefore, ICI therapy is recognized as a promising treatment method for advanced HNSCC patients. In fact, the NCCN Head and Neck Cancer Guidelines recommend ICI alone or in combination with chemotherapy for recurrent, distant metastatic, and cisplatin-refractory patients [23].

The problem with this ICI therapy is that it is associated with adverse events. Whereas conventional chemotherapy/radiotherapy for HNSCC induces treatment-related adverse events mainly due to its ability to induce cell death, the ICIs reduce immune tolerance and induce immune-related adverse events (irAEs) [24]. The time of onset and healing process are different from those by conventional chemotherapy. Although ICI-induced irAEs appear in all organs throughout the body, the skin, endocrine organs, digestive tract, liver, and respiratory organs are often affected [25]. In the oral cavity, there are many autoimmune diseases including Sjogren syndrome, rheumatoid arthritis of the temporomandibular joint, systemic lupus erythematosus (SLE), systemic sclerosis, Bechet’s disease, pemphigus vulgaris, pemphigoid, and oral lichen planus (OLP) [26,27]. Treatment of various malignancies, including HNSCC, with ICIs may result in the appearance of oral irAEs, but not all autoimmune diseases have been reported in patients treated with ICIs. Updating the knowledge of ICI-induced oral irAEs is essential for surgical oncologists and medical oncologists treating HNSCC and oral healthcare professionals treating oral disorders. In this article, we review the involvement of CTLA-4 and PD-1/PD-1 in tumor immunity, the mechanism by which anti-CTLA-4, anti-PD-1, and anti-PD-L1 antibodies exhibit anti-tumor effects and induce irAEs, the types of irAEs in ICI-treated HNSCC patients, and ICI-induced Sicca syndrome, oral lichenoid reaction (OLR), and oral pemphigoid lesions.

## 2. Mechanisms by Which ICIs Exert Antitumor Activity and Induce irAEs

### 2.1. Involvement of CTLA-4 and PD-1/PD-L1 in Cellular and Humoral Tumor Immunity

The CD8^+^ T cell is the main component of cell-mediated immunity against cancer. For activation of antitumor CD8^+^ T cells, tumor antigen needs to be presented by antigen-presenting cell (APC) such as dendritic cell (DC) and recognized by the T cell receptor (TCR) of CD8^+^ T cells. In addition, co-stimulatory signals are required for T cell activation. The co-stimulatory and co-inhibitory signals of T cells are mediated by the ligand–receptor pairs on the cell membrane [28]. The ligand–receptor pairs such as CD80/86-CD28, CD40L-CD40, and ICOSL-ICOS promote co-stimulatory signals between DC and T cells. The pairs of co-inhibitory molecules include CD80/86–CTLA-4, PD-L1/2–PD-1, BTLA–HVEM, MHCII–LAG-3, CD155/CD112/CD113–TIGIT, and galectin-9/phosphatidylserine–TIM-3 [13]. Current ICI therapy is mainly designed to suppress the action of co-inhibitory molecules to restore the anti-tumor capability of T cells.

When tumor antigens are released into the tumor microenvironment from dying tumor cells, antigen-presenting cells (APCs) such as DCs take them up by endocytosis and migrate to local lymph nodes. In endosomes of DCs, protein antigens are processed to peptide antigens and presented on the cell surface with HMC Class II, and recognized by CD4^+^ helper T cells. In contrast, tumor antigens synthesized inside cells are decomposed into peptide fragments by proteasomes, transported into the endoplasmic reticulum by the protein transporter (TAP), binding with MHC class I molecules, and are presented on the cell surface via the Golgi apparatus to be recognized by TCR of CD8^+^ T cells [24,28]. However, even when tumor antigens are taken up by DC from the outside, they can be degraded by the proteasomes, bind to MHC class I molecules, and cross-presented on the cell surface to be recognized by CD8^+^ T cells. In the lymph node, naive CD8^+^ T cells that receive tumor antigens must be activated by the co-stimulatory signal through binding CD80/86 of DC cells to CD28 of T cells to become cytotoxic CD8^+^ T cells. Naive CD4^+^ T cells that receive tumor antigens through the MHC class II pathway and co-stimulatory signals from DCs differentiate into T helper 1 (Th1) cells, T helper 2 (Th2) cells, T helper 17 (Th17) cells or regulatory T (Treg) cells with the use of corresponding cytokines required for each CD4^+^ T cell subtype [29,30]. Activated CD8^+^ T cells will express the co-inhibitory molecule CTLA-4 that can bind to CD80/86 of DCs more efficiently than CD28 and suppresses the T cell activity. This prevents excess activation of CD8^+^ T cells through DC stimulating signals [31]. After activation, CD8^+^ T cells move to a peripheral tumor site and recognize tumor peptide antigens presented via MHC class I and exert antitumor activity. However, if tumor cells express the co-inhibitory molecule PD-L1, PD-1 on T cells bind to PD-L1 of tumor cells and the cytotoxicity of CD8^+^ T cells will be reduced (Figure 1A). Treg cells, which constitutively express CTLA-4, suppress CD8^+^ T cells and CD4^+^ helper cells by inhibiting DC maturation, consuming IL-2 by their IL-2R, and producing the inhibitory cytokines TGF-β, IL-10, and IL-35 [32].

The germinal center (GC) plays an important role in the proliferation and differentiation of B cells. B cell activation usually requires signals from B cell receptors (BCR), B cell co-receptors, and CD4^+^ helper T cells [33]. As an APC, B cells detect protein antigens suitable for them using a BCR with a cell membrane-bound immunoglobulin (Ig). Thereafter, tumor antigens are incorporated inside cells, processed to peptide antigens, and then presented via HMC class II to T follicular helper (Tfh) cells in GC. Tfh and GC B cells are physically connected by ligand–receptor pairs such as CD80/86–CD28, CD40L–CD40, ICOS–ICOSL, and PD-L1–PD-1. Co-stimulatory/co-inhibitory signals are transmitted through these paired molecules [34,35]. When GC B cells meet antigen-specific Tfh cells, they activate each other [36,37]. After stimulation, GC B cells differentiate into plasma cells to produce secretory immunoglobulin as an antibody (Figure 1B) and also differentiate into memory B cells [33,37] (Figure 1B). IL-21 secreted from Tfh cells binds IL-21R, a co-receptor of GC B cells. IL-21R signaling can induce Bcl-6 expression and promote B-cell proliferation [38]. In contrast, T follicular regulatory (Tfr) cells suppress Tfh and B cells through CTLA-4 in GC. Production of inhibitory cytokines such as IL-10 and TGF-β may be another way by which Tfr cells inhibit B cell responses [39].

In melanoma patients, induction of tumor immunity is known to cause autoimmune disease [40]. Indeed, vitiligo was reported to occur as an autoimmune response in patients whose melanoma showed spontaneous remission. It is considered that this is because cytotoxic CD8^+^ T cells recognize both normal melanocytes, the target in this autoimmune disease, as well as melanoma cells and destroy these cells (Figure 1A). In animal experiments, white hair grew in black-haired mice after treatment with an antibody against tyrosinase and related proteins (TRP-1), a melanogenic enzyme. This antibody also blocked the growth of transplanted melanoma in mice, suggesting that melanocyte TRP-1 acts as tumor antigen [41]. Autoantibodies that recognize tyrosinase, TRP-1, and TRP-2, have been detected in melanoma patients with vitiligo [42].

### 2.2. Anti-CTLA-4, anti-PD-1, anti-PD-L1 Antibodies as Antitumor Agents and Autoimmune Disease Inducers

IrAEs have been reported to occur more often in patients treated with anti-CTLA-4 compared to patients treated with anti-PD-1 and anti-PD-L1 [43]. Among the ICIs currently used in HNSCC, ipilimumab against CTLA-4 and atezolizumab and durvalumab against PD-L1 are IgG1, while nivolumab and pembrolizumab against PD-1 are IgG4. In the case of IgG1 antibody, the Fc portion of antibody in immunocomplexes can mediate antibody-dependent cellular cytotoxicity (ADCC), antibody-dependent cellular phagocytosis (ADCP), and complement-dependent cytotoxicity (CDC) [44,45,46]. These actions may contribute to IgG1-mediated cytotoxicity.

When cancer patients are treated with anti-CTLA-4 antibodies, CD8^+^ T cells are released from their anergy state and attack tumor cells, but at the same time, CD8^+^ T cells also attack normal cells that present cross-reactive antigens. The CD4^+^ T cell is also released from a suppressive state [47]. When peripheral blood mononuclear cells (PBMCs) from melanoma patients who received a CTLA-4 blockade, tremelimumab, were activated in vitro, there was no difference in the levels of Th17 between patients with or without antitumor effects, but a marked increase in IL-17 and Th17 cells was observed in patients with inflammatory and autoimmune responses [48]. It was also reported that Fc receptor-positive phagocytes recognized the anti-CTLA-4 antibody conjugated to CTLA-4 on Treg and reduced the number of Treg [49,50]. In fact, in a mouse colorectal cancer model, anti-CTLA-4 antibodies caused a rapid decrease of intratumoral Tregs, while they increased peripheral Treg cells [51]. Similarly, treatment of colorectal cancer-bearing mice with anti-CTLA-4 antibodies decreased Foxp3^+^/CD4^+^ Treg cells, but increased CD4^+^ T and CD8^+^ T cells and the expression levels of pro-inflammatory Th1/M1-related cytokines IFN-γ, IL-1α, IL-2, and IL-12 [52]. The effects of anti-CTLA-4 antibody on CD8^+^ T, CD4^+^ T, and Treg cells may contribute to the induction of autoimmune diseases.

PD-1 plays an important role in maintaining peripheral immune tolerance. The ligand PD-L1 is constitutively expressed on B cells, T cells, macrophages, and DCs, while PD-L2 is not expressed on quiescent cells and is induced by pro-inflammatory cytokines [32]. Anti-PD-1 antibody inhibits the binding of PD-1 and PD-L1/L2 on tumor cells. As a result, cytotoxic CD8^+^ T cells exert efficient antitumor activity via Fas, granzyme B, perforin, and IFN-γ. In fact, an increase in the levels of activated effector memory T cells and central memory T cell subsets of CD4^+^ and CD8^+^ T cells, and activated Th1 plus T-helper follicular 1 cells was observed in melanoma patients treated with anti-PD-1 antibodies [53]. The relationship between PD-1 and Treg cells was investigated in animal models where anti-PD-1 antibody suppressed tumor growth and prolonged survival of osteosarcoma cancer-bearing mice. Anti-PD-1 antibodies were found to decrease intratumoral Treg cells, while they increased tumor-infiltrating immune cells [54]. In a chronic inflammation model, linking PD-1 on Treg cells to PD-L1 on CD8^+^ T cells suppressed the activity of CD8^+^ T cells, which was abolished by reducing PD-1 expression of Treg cells [55]. In contrast, Toor et al. reported that pembrolizumab efficiently blocked PD-1 expression on peripheral blood cells of healthy subjects and breast cancer patients in vitro, but that it did not affect the expression of Treg-related markers or function. The pembrolizumab-mediated blockade was observed in only CD4^+^CD25^+^ non-Treg cells [56]. Kamada-T et al. [57] reported patients with hyperprogressive disease in whom anti-PD-1 had no antitumor effects but rather caused rapid tumor growth with the proliferation of PD-1^+^ Treg, suggesting that the anti-PD1 antibody can induce a strong immunosuppressive status. In patients who receive anti-PD-1 antibodies and develop the hyperprogressive disease, an attempt is made to suppress Treg cells by adding anti-CTLA-4 antibodies [58].

A characteristic of autoimmune diseases is the production of autoantibodies, where B cells play a major role [59]. Autoantibodies against thyroid antigens are detectable in ICI-related thyroiditis and hypothyroidism [60]. Autoantibodies against islet cell antigens and glutamic acid decardoxylase-65 were observed in patients who developed diabetes after ICI treatment [61,62]. Pemphigoid observed in ICI-treated patients is associated with the production of autoantibodies against basement membrane zone antigens [63,64]. It has been shown that a blockade of PD-1/PD-L1 signal can either enhance or suppress the humoral immune response [37]. In a mouse model, blockage of the PD-1 blockade was found to augment humeral immunity through the accumulation of GC CD4^+^ T cells expressing ICOS and differentiation of B cells [33]. Anti-PD-1 antibody may exert its enhancing effect on B cells through Tfh and Tfr, but it can also directly act on PD-1 expressed on B cells [37]. Das et al. [65] reported that although anti-CTLA-4 and anti-PD-1 antibodies had no apparent effects on B cells in advanced melanoma patients when administered individually, their combination increased in serum plasmablasts and CXCL13, a marker of GC activation. Particularly, in a subset of CD21^low^ memory B cells, the combined treatment decreased the expression of PD-1, and increased the number of cells showing high IFN-γ signals, indicating the ability of the anti-PD-1 antibody to promote humoral immunity. In contrast, B cell reactivity elicited by protein immunization was shown to be reduced in mice that lack PD-1 and PD-1 ligands [33,66,67]. The anti-PD-1 antibodies may weaken the binding integrity of Tfh cells with GC B cells through CD40-CD40L and ICOS-ICOSL and prevent the activation of B cells (Figure 1B). Willsmore et al. [59] stated that although B cells express PD-1 and PD-L1, these are not recognized as primary targets for anti-PD-1 antibody therapy.

ICIs treatment causes relapse in patients with autoimmune disease [68]. As antigen epitopes are shared with the tumor and target organ of autoimmune diseases, ICIs will lower the threshold for antigen-specific T cell activation and cause irAEs [69]. Autoantibodies were reported to become detectable in 19.2% of melanoma patients after treatment with anti-CTLA-4 antibody [70]. Relapse of autoimmune disease after ICI treatment was also observed in 38% of melanoma patients [71]. This suggests the importance of pre- or subclinical autoimmune disease in the onset of irAEs but also indicates the role of other factors in the development of ICI-induced irAEs. The direct effect of ICIs on target cells has also been reported. A typical event is hypophysitis. This is because CTLA-4 is highly expressed in the pituitary gland, so anti-CTLA-4 antibodies directly bind and injure the gland through ADCP and/or CDC to induce hypophysitis [44]. PD-L1 is expressed not only in lymphoid tissues but also in non-lymphoid tissues such as vascular endothelial cells, the thyroid, muscle, liver, placenta, mesenchymal cells, and the pancreas to maintain peripheral immune tolerance [72,73,74,75]. Anti-PD-L1 antibodies may bind directly to cells of these tissues/organs and exhibit cytotoxicity via Fc receptors.

Bacterial species such as *Bacteroides, Clostridium,* and *Faecalibacterium* were shown to expand Treg and produce anti-inflammatory cytokines. Recently, the relationship between the gut microbiota and irAEs has been investigated [76,77]. Dubin et al. [76] found that species belonging to the Bacteroidetes phylum conferred resistance to colitis that developed after anti-CTLA-4 antibody therapy, suggesting that expansion of these bacteria helps to prevent the development of ICI-induced colitis. The microbiota plays an important role in the endogenous synthesis of water-soluble B vitamins. A reduced capacity for microbe-mediated production of B vitamins and polyamine transport may lower the threshold for the onset of immune-mediated colitis. The group 3 innate lymphoid cells (ILC3s) are the major innate lymphoid cell type involved in the immunopathology of ICI-induced colitis. *Lactobacillus reuteri* (*L.reuteri*) administration markedly decreased the mucosal numbers of ILC3s and therapeutically prevented colitis in the ICI-treated mice. This suggests that L. reuteri affects the immunopathology of ICI-associated colitis primarily by altering the local number of ILC3s. [78]. Oral administration of Bacteroides fragilis, *Bacterodes thetaiotaomicron,* and *Burkholderia cepacia* was found to restore the antitumor effects of anti-CTLA-4 antibody in MCA205 sarcoma-bearing mice treated with a broad-spectrum antibacterial cocktail. These bacteria enhanced antitumor immunity by inducing the Th1 immune response in regional lymph nodes and promoting DC maturation [79]. Some oral bacteria migrate to the intestinal tract. Therefore, oral bacteria may be useful to alleviate the colitis caused by ICIs [80].

## 3. ICI-Induced irAEs in Advanced HNSCC Patients

In HNSCC, PD-L1 is highly expressed in a variety of immune and non-immune cells, including tumor cells and cancer-associated fibroblasts [81]. As advanced HNSCC patients are frequently treated with anti-PD-1 antibodies, most reported irAEs are related to nivolumab and pembrolizumab [17,18,19,20,21,22,82,83,84] (Table 1). A study by Ferris et al. compared 361 HNSCC patients treated with nivolumab to patients treated with standard chemotherapy including methotrexate, docetaxel, or cetuximab. Mean survival was 7.5 and 5.1 months, respectively, overall survival was significantly increased in the nivolumab group and 1-year survival was 36 and 16.6%, respectively, improving by 19%. The overall incidence of irAEs induced by nivolumab was 58.9% (139/236) and that of grade 3–4 was 13.1% (31/236), while the incidences of overall and high-grade AEs due to standard chemotherapy were 77.5% (86/111) and 35.1% (39/111), respectively; the incidence of irAE by nivolumab is low compared with conventional chemotherapy. The most common ICI-induced irAEs were gastrointestinal disorders that occurred in 61 (25.8%) patients, of which 20 (8.5%) involved nausea, 17 (7.2%) appetite decrease, 16 (6.8%) diarrhea, and 8 (3.4%) vomiting. Dermatological irAEs occurred in 42 (18%) patients, of which 18 (7.6%) involved rash, 17 (7.2%) pruritus, and 7 (3.0%) dry skin. As a systemic event, fatigue was observed in 33 (14.0%) patients (Table 1). Matsumo et al. [22] reported endocrine disease as an irAE by nivolumab in 14 (26.4%) of 108 patients. Bauml et al. [19] analyzed patients treated with pembrolizumab, and reported that the overall incidence of irAE was 63.7% (109/171) and that of grade 3–4 irAEs was 15.2% (26/171). Of all irAEs, 30 (17.5%) involved gastrointestinal disorders, 21 (12.3%) dermatologic disease, 27 (15.8%) liver dysfunction, 30 (17.5%) general disorders, and 16 (9.3%) endocrine diseases. The frequency of serious irAEs was low. Oral mucosal diseases were reported by Mehra et al. [83] in 4 (2.1%) of 192 patients and by Burtness et al. [21] in 9 (3%) of 300 patients without serious irAEs.

Compared with HNSCC, ICI treatment for melanoma (CheckMate 037) was started in advance, in which the overall frequency of nivolumab-induced irAEs was 59% (157/268) and that of grade 3–4 was 9% (22/268) (Checkmate 037 2015) [16]. In another study of 278 melanoma patients, pembrolizumab (Check Mate 006) induced irAEs in 221 (79.5%) patients and grade 3–5 occurred in 37 (13.3%) patients (Keynote 006 2015) [15]. In 418 lung cancer patients treated with nivolumab, (Checkmate 057), irAEs occurred in 283 (68%) patients and grade 3–4 occurred in 44 (10%) patients [85]. The frequencies of overall irAEs in ICI-treated HNSCC patients ranged from 30 to 69.6% (Table 1). Therefore, there was no apparent difference in the incidences of nivolumab-induced irAE among patients with melanoma, lung cancer, and HNSCC. The major organs involved in these malignancies are the digestive tracts and skin.

Recently, the anti-PD-L1 antibodies durvalumab and atezolizumab have been used for the treatment of HNSCC patients [46,86]. Sui et al. [46] treated advanced HNSCC patients with the anti-PD-L1 antibody durvalumab and/or anti-CTLA-4 antibody tremelimumab. The overall irAE incidences in patients treated with durvalumab, tremelimumab, and their combination were 63.1% (41/65), 55.4% (36/65), and 57.9% (77/133), respectively, and those of grade 3–4 irAEs were 12.3% (8/65), 16.9% (11/65), and 15.8% (21/133), respectively. These antibodies against PD-L1 induced irAEs in HNSCC patients at frequencies similar to nivolumab and pembrolizumab.

## 4. Induction of Sicca syndrome by ICIs

Primary Sjogren syndrome (pSS) is a systemic autoimmune disease in which symptoms appear in many exocrine organs during the course of the disease with diverse clinical features, but in principle salivary and lacrimal glands are affected and dry mouth and dry eye occur. Many factors, including environmental and genetic factors, play roles in the development of pSS [87,88]. In histology of the affected exocrine glands, there is periductal accumulation of mononuclear cells composed of mainly CD4^+^ T and B cells. GC formation, hypergamma globulinemia, and the production of autoantibodies SS-A and SS-B reflect the excessive activity of B cells. Among the CD4^+^ T cells that accumulate periductally, Th1, Tfh, and Th17 are implicated to play important roles in the development of pSS [89,90,91].

There are case reports of Sjogren syndrome/Sicca syndrome that occurred in association with ICI treatment for various malignancies with similar clinical features as pSS that have been published. According to the reports, dry mouth and dry eyes were observed in 98 (5.3%) of 1832 patients. The frequency of Sicca syndrome observed with the combination of anti-CTLA-4 antibody and anti-PD-1 antibody was 9.4%, and that with anti-CTLA-4 alone was 1.4% [92]. Le Burel et al. [93] reported that out of 908 patients who received anti-PD-1/PD-L1 treatment, the estimated prevalence of Sjogren syndrome was 0.3%, while it increased to 2.5% among patients treated with the combined use of ICIs. Ramos-Casals et al. [92] summarized the characteristics of their 26 patients who developed ICI-induced Sicca/Sjogren syndrome. The characteristics of the reported 76 patients, including sex distribution, underlying malignancies, time to onset, prevalence of pSS-associated autoantibodies, other irAEs, and degree of improvement of irAEs, were analyzed [92,93,94,95,96,97,98,99,100] (Figure 2). pSS has been shown to occur predominantly in women, with 95% of patients reported to be female [92], but among the 76 patients with Sicca syndrome caused by ICIs, 48 (63%) patients were male and 28 (37%) patients were female (Figure 2A). In the study by Ramos-Casals et al. [92], the male–female ratio was 1:1. Therefore, Sicca syndrome due to ICIs is not confined to women. The primary malignancies were melanoma, lung cancer, and renal cancer (Figure 2B). Melanoma accounted for 38% of the total. Oral cancer was limited to two cases.

Of the 52 patients in whom the names of antibodies were clarified, nivolumab, pembrolizumab, and their combination were administered to 18 (34.6%), 10 (19.2%), and 10 (19.2%) patients, respectively. After the start of ICI treatment, Sicca syndrome occurred at a constant rate every month, with 71% by half a year and 92% by 12 months (Figure 2C). A total of 37/76 (47%) patients developed Sicca syndrome by 3 months. In the case of pSS, 70% of immune cells that infiltrated into the salivary gland were shown to be T cells, of which 50% were CD4^+^ T cells and 20% was CD8^+^ T cells; the proportion of B cells was 22% [101,102]. In a study on ICI-induced Sicca syndrome, most infiltrating cells were T cells, with more CD4^+^ T cells than CD8^+^ T cells and only a few CD20^+^ B cells [98]. A histological study of the lower lip mucosa revealed that although aggregated CD20^+^ B cells and CD3^+^ T cells were observed, T cells were predominant and the CD8^+^ T cell was more frequent than the CD4^+^ T cell [100]. This indicates that much lower levels of B cells infiltrate the salivary gland in ICI-induced Sicca syndrome compared with pSS. In a previous study analyzing 26 patients, there was a prevalence of SS-associated serum autoantibodies (48% anti-nuclear antibody (ANA), 32% rheumatoid factor (RF), 48% SS-A, and 25% SS-B) [92]. In 72 patients in this study, the detection rates of ANA, RF, SS-A, and SS-B were 42, 14, 17, and 12%, respectively (Figure 2D). In the case of pSS, these antibodies were detectable in 79, 48, 73, and 45%, respectively. The prevalence of serum autoantibodies in ICI-induced Sicca syndrome is much lower in comparison with pSS. Other irAEs observed in the patients with ICI-induced Sicca syndrome include endocrine, dermatological, musculoskeletal, digestive, and liver disorders (Figure 2E).

When the symptoms of irAEs are severe, ICI administration is discontinued. Indeed, 36/65 (55%) discontinued ICI treatment due to irAE. For the treatment of ICI-induced Sicca syndrome, systemic corticosteroid was administered. As a result, among 31 evaluable patients, significant improvement was observed in 9 (29%), moderate improvement in 9 (29%), mild improvement in 10 (32%), and no improvement in 3 (10%) patients (Figure 2F). Corticosteroids were demonstrated to be useful, but their therapeutic effects were insufficient in 12 (42%) patients.

## 5. Induction of Oral Mucosal Lesions by ICIs

The frequency of skin irAEs due to anti-PD-1 antibody was relatively high, with rash at 7.2%, pruritus 7.2%, and dry skin at 3%. When patients with HNSCC were treated with anti-PD-1 antibody, the incidence of stomatitis was 2.1% and that of grade 3–4 disease was 0.4%, which is less frequent than skin disorders caused by ICIs [17]. With conventional chemotherapy for HNSCC, 9% of patients treated with cisplatin + fluorouracil developed mouth ulcer, of which 2.7% was grade 3–4 [17]. Therefore, the frequency of mouth ulcer caused by ICIs is also low compared with conventional chemotherapy.

OLP is a chronic inflammatory mucosal disease for which the cause of the autoimmune reaction has not been identified. It frequently occurs in women and involves the formation of reticular lesions on the buccal mucosa on both sides. Histologically, it is characterized by chronic inflammation and submucosal band-shaped T cell infiltration [103,104]. In contrast, an oral lichenoid lesion (OLL), with similar clinical and histological features to OLP, is associated with dental restorations, drugs, liver disease, hypertension, systemic diseases such as diabetes, and food and spice allergies. Therefore, OLL can be further divided into four types: amalgam restoration-associated OLL; drug-related OLL; OLL in chronic graft-versus-host disease (GVHD); and OLL unclassified [105,106]. OLL is also called OLR, lichenoid stomatitis, and lichen planus-like lesion [105]. In this review, OLP-like lesions were called OLR [107,108,109,110,111,112,113]. In the oral cavity, ICI-induced OLR affects the buccal mucosa, tongue, and gums and shows clinical appearances including reticular, white-lined, ulcer, erythema multiforme (EM)-like, and GVHD-like lesions. Since no obvious symptoms were observed in most white plaque and papular lesions, biopsy was not necessarily performed.

Indeed, of the 32 patients with OLR lesions reported previously, only 11 (34%) underwent histological examination. Careful clinical examination is required to discover oral lesions in patients with no or mild symptoms. Of the 32 reported patients with ICI-related OLR, 20 patients were treated with nivolumab, 8 patients with pembrolizumab, and 4 patients with atezolizumab. As the primary malignancies, 7 lung cancers, 7 melanomas, 4 renal cell carcinomas, and 4 OSCCs were included (Table 2). The onset time ranged from 1 to 23 cycles. Histological examination of ulcerative lesions revealed a total epithelial defect and subepithelial lichen planus-like lymphocyte infiltration. Eosinophil or neutrophil infiltration is not usually observed in OLP [114]. Shazib et al. [112] reported 4 cases of lichenoid mucositis with eosinophilia, suggesting that the mucosal lesion is not OLP, but has a histological characteristic of drug hypersensitivity. Immunoreactive self-antigen was not detected in these OLR lesions by the direct immunofluorescence (DIF) method.

Outside the oral cavity, skin papules, skin LPs, and pneumonia are observed as irAEs. Most patients with OLR were treated with topical application of corticosteroids. When the effect was insufficient, systemic corticosteroids were administered. All of them improved after several weeks. ICIs were discontinued in 10/18 (56%) patients due to irAE. Shi et al. [107] found that 16/20 (80%) of ICI-treated cancer patients had already taken drugs with a potential to induce OLR, and suggested that ICIs broke immune tolerance locally and evoked an immunological response to these drugs. On the other hand, a 43-year-old woman with follicular lymphoma was treated with rituximab and developed OLR [115]. A 68-year-old woman also developed OLR with oral pain caused by abatacept on treating rheumatoid arthritis [116]. Abatacept is CTLA-4-Ig that binds CD80/86 on DC and blocks the co-stimulatory signal through CD28 of T cells, which suppresses T cell-mediated inflammation of rheumatoid arthritis. These findings suggest that OLR may occur as hypersensitivity to the monoclonal antibody in these patients.

Severe oral ulcers due to ICIs have been also reported, and some extend to the pharynx [117,118,119,120]. In three patients with severe ulcers, ICI discontinuation and systemic administration of prednisolone improved the lesions. A 93-year-old patient received nivolumab for the treatment of pharyngeal cancer and developed multiple oral ulcers during the treatment, while the tumor disappeared. The ulcers were not improved by oral prednisolone or intravenous methylprednisolone, but improved by oral cyclophosphamide and colchicine gargle. This suggests that the ulcers are associated with Behcet’s disease [119].

In addition, there are patients who developed oral pemphigoid lesions related to ICI treatment [63,121,122,123,124,125,126,127,128]. ICI-induced mucous membrane pemphigoid (MMP) is rare, with only two patients being reported [63,125]. MMP patients have autoantibodies to the basement membrane zone antigens, and the main target of autoantibodies is the hemoidesmosomal protein BP180. MMP lesions caused by ICIs appeared on the gingiva, tongue, and buccal mucosa without other systemic lesions. Direct immunofluorescence (DIF) was positive. The MMP lesions improved by discontinuation of ICIs and administration of doxycycline. Nine patients with bullous pemphigoid (BP) with oral lesions have been reported (Table 2). Zumelzu et al. [63] examined the characteristics of 29 patients with ICI-induced BP, of whom 6 (21%) patients had oral lesions. Sadik et al. [127] reported oral involvement in 2/12 (17%) BP patients. Oral BP lesions appear on the buccal mucosa, palate, and tongue. The major self-antigens are BP180 and BP230 of the dermal–epidermal junction [128]. BP180 is known to be expressed on the cell surface of melanoma, non-small cell lung cancer, and urothelial epithelium. Administration of anti-PD-1/PD-L1 antibody will promote immune cross-reactivity against BP180 on tumor cells and on the basement membrane of the skin, leading to the development of BP [129]. Because a high level of BP180 expression in NSCLC tissue is correlated with a better therapeutic effect of ICIs, serum BP180 is a potential marker to infer autoantibody production and responsiveness to ICI therapy [64]. Yatim et al. [130] reported a patient with paraneoplastic pemphigus and detected Dsg1 and Dsg3 by serological examination. Discontinuation of ICIs, topical dexamethasone (DEX), oral prednisolone, and doxycycline improved the lesions of BP and paraneoplastic pemphigus.

## 6. Conclusions

With the use of ICIs for the treatment of HNSCC, development of a new autoimmune disease called irAEs has become a problem. The mechanisms include the disruption of immune tolerance in cellular and humoral immunity, enhanced cross-reactivity between tumor antigens and self-antigens on target organs of autoimmune disease, exacerbation of pre-existing autoimmune reactions, and Fc component-mediated cytotoxicty. ICI treatment of HNSCC patients induces various irAEs including dermatological, gastrointestinal, and endocrine disorders. Treatment of various malignancies, including HNSCC, with ICIs may result in the appearance of oral irAEs. Major oral manifestations of irAEs include Sicca syndrome, OLR, and pemphigoid. They are mostly mild, but ICI treatment is frequently discontinued and systemic corticosteroids are required to control the irAEs. Unlike pSS, ICI-induced salivary gland lesions have a lower frequency of B cells. ICI-induced OLRs may be more closely correlated with drug allergies. This suggests that these irAEs are formed by mechanisms different from pSS and OLP. For successful treatment of HNSCC with ICIs, it is necessary to elucidate the mechanisms by which irAEs occur and develop a diagnostic method and treatment for ICI-induced lesions. Among sociological factors, gender and race had a positive impact on the effectiveness of ICI by increasing effector T cell activity and PD-1 expression. Aging, on the other hand, impaired responsiveness to ICI by reducing the number of functional T cells and increasing toxicity [131]. It is also required to investigate how these sociological factors affect the development of irAE in the oral cavity.

## Figures and Tables

**Figure 1 cancers-14-00792-f001:**
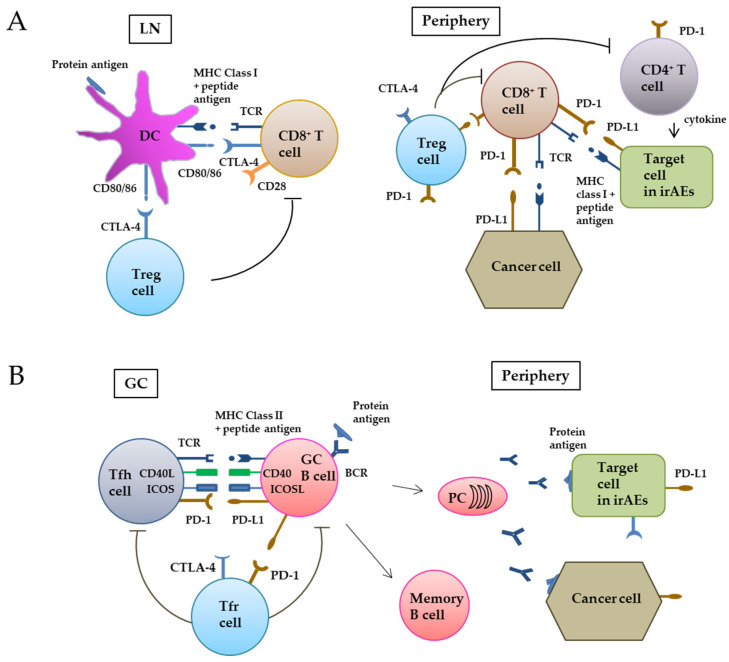
CTLA-4 and PD-1/PD-L1 in cellular (**A**) and humoral (**B**) tumor immunity. (**A**) In the CD8^+^ T cell-based pathway, when tumor antigens are released into the tumor microenvironment from dying tumor cells, DCs take up the protein antigens and decomposes them to peptide antigens. In regional lymph nodes, the peptide antigens are cross-presented via MHC class I on the cell surface and recognized by the TCR of CD8^+^ T cells. In addition, the co-stimulatory binding of CD80/86 of DCs with CD28 of T cells is required for naïve CD8^+^ T cells to differentiate into cytotoxic CD8^+^ T cells. Activated CD8^+^ T cells express the co-inhibitory molecule CTLA-4 to prevent excess activation of CD8^+^ T cells. CD8^+^ T cells move to a peripheral tumor site and recognize tumor peptide antigens presented via MHC class I and exert antitumor activity. However, PD-I on CD8^+^ T cells binds PD-L1 on tumor cells and suppresses the activity of CD8^+^ T cells. Treg cells locally suppress the activity of CD8^+^ T cells and CD4^+^ helper cells. (**B**) The germinal center (GC) plays an important role in the proliferation and differentiation of B cells. Tfh cells physically bind to GC B cells by co-stimulatory/co-inhibitory pairs such as CD28–CD80/86, CD40L–CD40, ICOS–ICOSL, PD-1, and PD-L1. BCR expressed on B cells detects the protein antigens, and take them into the cells. Peptide antigens processed in B cells are then presented via MHC class II and recognized by Tfh cells. Interaction between Tfh and B cells promotes the differentiation of GC B cells into plasma cells and memory B cells. Tfr cells suppress Tfh and B cells. LN, lymph node; Treg, regulatory T; TCR, T cell receptor; irAEs, immune-related adverse events; GC, germinal center; Tfh, T follicular helper; Tfr, T follicular regulatory; PC, plasma cell; BCR, B cell receptor.

**Figure 2 cancers-14-00792-f002:**
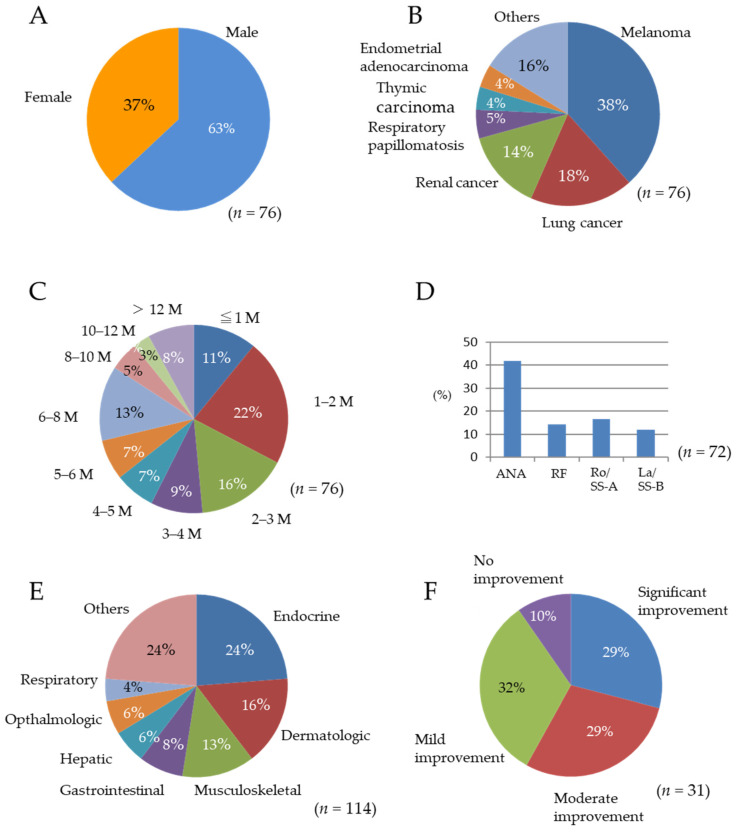
Characteristics of 76 previously reported patients with ICI-induced Sicca syndrome. (**A**) Sex. (**B**) Types of underlying malignancies. (**C**) Time to onset, months. (**D**) Prevalence of Sicca syndrome-related serum autoantibodies. (**E**) Other irAEs. (**F**) Degree of improvement of irAEs.

**Table 1 cancers-14-00792-t001:** Immune checkpoint inhibitor-induced adverse events in studies of HNSCC patients.

ICIs	First Author	Any Event		Dermatological		Endorine		Gastro-Intestinal		Hepatic		Pulmonary		General		Oral	
	[Ref.]	AG (%)	G3 ≤(%)	AG (%)	G3 ≤(%)	AG (%)	G3 ≤(%)	AG (%)	G3 ≤(%)	AG (%)	G3 ≤(%)	AG (%)	G3 ≤(%)	AG (%)	G3 ≤(%)	AG (%)	G3 ≤(%)
Nivolu mab	Ferris	139 (58.9)	31 (13.1)	42 (18.0)	0			61 (25.8)	0					37 (15.7)	5 (2.1)	5 (2.1)	1 (0.4)
	[17] n = 236			Rash				Nausea						Fatigue	Fatigue	Stomatitis	
				Pruritus				Appetite decrease						Asthenia			Stomati tis
				Dry skin				Diarrhea									
				Alopecia				Vomiting									
	Kiyota	16 (69.6)	2 (8.7)	8 (34.8)	0			8 (34.8)	0					4 (17.4)	0	1 (4.3)	0
	[82] *n* = 23			Rash				Nausea						Fatigue		Stomatitis	
				Pruritus				Appetite decrease									
				Alopecia				Diarrhea									
	Okamoto	30 (30)	NR	3 (3)	1 (1)	17 (17)	0	2 (2)	1 (0.4)	4 (4)	2 (2)	11 (11)	3 (3)	1 (1)	0		
	[84] n=100			Dermatitis		Hypo thyroidism		Gastro- intestinal		Liver	Liver	Interstitial	Interstitial	Weight loss			
					Dermatitis	Hyper thyroidism		Hemorrhage	Hemorrhage	Dysfunc tion	Dysfunc tion	lung	lung				
						Adrenal insufficiency		Diarrhea				disease	disease				
	Matsuo	53 (49.1)	11 (10.2)	11 (20.8)	1 (1.9)	14 (26.4)	2 (3.8)	8 (15.1)	0	6 (11.4)	3 (5.7)	4 (7.6)	2 (3.8)	1 (1.9)	0		
	[22] *n* = 108			Rash/	Rash/	Hypo thyroidism		Diarrhea		Elevated hepatic	Elevated hepatic	Pneumonia	Pneumonia	Fever			
				Pruritus	Pruritus	Hypophysitis	Hypophysis	Nausea		enzymes	enzymes						
						Hyper glycemia	Hyper glycemia			Cholangitis							
Pembrolizumab	Bauml	109 (63.7)	26 (15.2)	21 (12.3)	1 (0.6)	16 (9.3)	0	30 (17.5)	1 (0.6)	27 (15.8)	6 (3.5)	11 (6.4)	2 (1.2)	30 (17.5)	1 (0.6)		
	[19] *n* = 171			Rash	Rash	Hypo thyroidism		Nausea		AST increase	AST increase	Pneumoni tis	Pneumonia	Fatigue	Fatigue		
				Pruritus				Diarrhea	Diarrhea	ALT increase	ALT increase	Cough					
								Appetite decrease		Bilirubin increase	Bilirubin increase						
										ALP increase	ALP increase						
	Mehra	123 (64)	24 (12.5)	47 (24.5)	NR	23 (12)	2 (1)	36 (18.8)	2 (1)	11 (5.7)	6 (3.1)	5 (2.6)	2 (1)	63 (32.8)	2 (1)	4 (2.1)	NR
	[83] *n* = 192			Rash		Hypo thyroidism	Hypo thyroidism	Appetite decrease	Appetite decrease	AST increase	AST increase	Pneumoni tis	Penumonia	Fatigue	Fatigue	Stomatitis	
				Pruritus		TSH level		Nausea		ALT increase	ALT increase			Pyrexia			
				Dry skin		increase		Diarrhea Vomiting						Weight loss			
	Burtness	164 (54.7)	NR	96 (32)	10 (3.3)	65 (21.7)	5 (1.7)	206 (68.7)	26 (8.7)			139 (44.7)	34 (11.3)	162 (54)	22 (7.3)	9 (3)	0
	[21] n-300			Rash	Rash	Hypo thyroidism	NR	Constipation	Constipation			Cough	NR	Fatigue	Fatigue	Stomatitis	
				Derma titis				Diarrhea	Diarrhea					Asthenia	Asthenia		
				acneiform				Vomiting	Vomiting					Pyrexia	Pyrexia		
								Appetite decrease									
Durvalu mab	Siu	42 (63.1)	8 (12.3)	6 (9.2)	0	7 (10.8)	0	11 (16.9)	0					17 (26.2)	2 (3.1)		
	[46] *n* = 65			Rash				Diarrhea						Fatigue	Fatigue		
				Pruritus				Appetite decrease						Asthenia			
								Nausea Vomiting									
Tremelimumab	Siu	36 (55.4)	11 (16.9)	8 (12.3)	0	1 (1.5)	0	23 (35.4)	3 (4.6)					12 (18.5)	1 (1.5)		
	[46] *n* = 65			Rash		Hypo thyroidism		Diarrhea	Diarrhea					Fatigue	Fatigue		
				Pruritus				Appetite decrease						Asthenia			
								Nausea Vomiting						Pyrexia			
Durvalu mab	Siu	77 (57.9)	21 (15.8)	14 (10.5)	0	11 (8.3)	0	39 (29.3)	4 (3.0)					27 (20.3)	4 (3.0)		
	[46] *n* = 133			Rash		Hypo thyroidism		Appetite decrease						Asthenia	Asthenia		
Tremelimumab				Pruritus				Nausea Vomiting						Fatigue	Fatigue		
								Diarrhea	Diarrhea					Pyrexia			

ICI, immune checkpoint inhibitor; AG, any grade; G, grade; NR, not reported.

**Table 2 cancers-14-00792-t002:** Immune checkpoint inhibitor-induced adverse events in the oral cavity.

First Author	Age/Sex	Cancer Type	ICIs	Time to irAE	Clinical Feature (Distribution)	Pathological	Immunological	Other irAEs	Tumor	Treatment of irAEs	Clinical Outcome
[Ref.]				M/cycle		Feature	Data		Response		
Oral lichenoid reaction											
Shi	F	RCC	Nivolumab	1.6 M	Papular (mouth)	Lichenoid		Papular (palms, soles)	SD	ICI continued	
[107]	F	Lung cancer	Nivolumab	10.2 M	Mucositis (mouth)	Lichenoid		None	PR	ICI continued	
	M	Melanoma	Nivolumab	0.5 M	Erosive lichen planus	Lichenoid		None	SD	ICI discontinued	
					(mouth, penis)						
	F	RCC	Atezolizumab	8.3 M	Papular (mouth)	Lichenoid		Papular (plams, arms)	PR	ICI discontinued	
Sibaud	53/M	Multiple myeloma	Nivolumab	2 cycles	Papule, Reticular streaks			Cutaneous lichenoid		DEX	Resolved after
[108]					(lip, tongue, buccal)			eruption		Mouth wash	several weeks
	62/M	RCC	Nivolumab	23 cycles	White streaks (buccal),			No dermal lesions		None	Resolved after ICI
					White plaque (tomhue)					ICI discontinued	discontinuation
	42/M	Glioblastoma	Nivolumab	2 cycles	White papule (lip, tongue,			No dermal lesions		Topical corticosteroids	
		multiforme			buccal)					Anti-fungal lozenges	
	70/F	Lung cancer	Nivolmab	6 cycles	White streaks (buccal, lip, mouth			Cutaneous lichenoid		Topical and oral	Resolved after
					floor, soft palate, tongue),			eruption		corticosteroids	several weeks
					Erythema/atrophy (tongue)			Pneumonitis			
	41/F	Breast cancer	Pembrolizu mab	10 cycles	White plaque-like lesion			No dermal lesions		None	
					(tongue)						
	63/M	Lung carcinoma	Nivolumab	3 cycles	Reticular white streaks			Nonspecifc maculo-		Topical coticosteroids	
					(buccal, soft palate)			papular rash			
	56/M	Renal cell	Atezolizumab	11 cycles	White plaque-like lesions			Cutaneous lichenoid		Topical corticosteroids	Resolved after
		carcinoma			(tongue), Reticular white			reaction		ICI discontinued	ICI discontinuation
					streaks (hard palate)						
	66/M	Adenocarcinoma	Atezolizumab	14 cycles	Reticular white streak			Xerostomia		None	
		of esophagus			(buccal)						
	54/M	RCC	Atezolizumab	5 cycles	Ulcers (floor of mouth)			No dermal lesions		Topical corticosteroids	Resolved eventually
					Sensitive tongue						
	58/M	Lung carcinoma	Pembrolizu mab	12 cycles	Reticular white streaks			Cutaneous		Topical corticosteroids	Resolved eventually
Oral lichenoid reaction											
Namiki	84/F	Melanoma	Nivolumab	3 M	Ulcers (buccal, tongure)	Epithelial necrosis,		None		Methylprednisolone,	Resolved after
[109]						Lichenoid lymphocyte				Oral prednisolone	a month
						infiltration					
Shazib	74/F	Melanoma	Nivolumab	4 doses	Lichenoid (buccal, lip,	Lichenoid mucositis		Pneumonitis	PR	Topical CLO	
[112]					gingiva)						
	55/F	Melanoma	Nivolumab	16 doses	Lichenoid (palate)			Pneumotitis	PR	Topical FLUOC	
	68/M	OSCC	Nivolumab	2 doses	Lichenoid (buccal, gingiva)	Lichenoid mucositis			PR	Topical DEX	
	39/M	OSCC	Nivolumab	2 doses	Oral erythema					Topical DEX	
					multiforme (bucal, lip, palate)					Prednisolne	
	60/F	Breast cancer	Pembrolizumab	2 doses	Lichenoid (tongue, buccal,					Topical DEX	
					gingiva, lip)					Prednisolone	
Severe immune mucositis											
Cardona	93/M	Pharyngeal carcinoma	Nivolumab	10 cycles	Ulcers (buccal, tongue,	Infiltration of		Hypothyroidism	CR	Topical triamcinolone	Complete resolution
[119]					50% of oral mucosa)	lymphocytes and		Dysphagia		Oral prednisolone	Cyclophosphamide
					silimar to GVHD,	macrophage-like cells				Methylprednisolone	and colchicine
					Behcet’s disease					Cyclophosphamide, Colchicine	
Acero Brand	69/M	Layngeal carcinoma	Pembrolizumab	14 cycles	Oral and pharynx	Ulcerative esphoagitis		None	CR	ICI discontinued	Marked improvement
[117]					mucositis, esphagitis	with granulation tissue				Methylprednisolone	within 48 h
Miyagawa	75/M	Gastric cancer	Nivolumab	19 cycles	Erosion and ulcers	Infiltration of band-like	Dsg1 (-), Dsg3 (-)	Perianal erosions		ICI discontinued	Improved gradually
[118]					(buccal mucosa, tongue, lip)	inflammatory cells	BP180 (-), BP230 (-)	Erosion (glans,		Pednisolone	
							IIF (-), DIF (-)	penis)			
Wang	32/M	Gastric cancer	Camrelizu mab	15 cycles	Behchet’s disease, Ulcers (lip,					ICI discontinued	Lip lesion healed
[120]					penis, abdominal, skin),					Oral prednisolone	after few days
					folliculitis/acne (hands, feet)					and thalidomide	
Oral lichenoid reaction											
Obara	67/M	Lung adenocarcinoma	Nivolumab	0.5 M	Ulcers (entire oral mucosa,	Epithelial necrosis,	Dsg1 (-), Dsg3 (-)	None	PD	Topical triamcinolone	Resolved afer
[106]					lip, tongue)	Lichenoid lymphocyte	BP180 (-)				3 weeks
						infiltration					
	74/F	Lung adenocarcinoma	Nivolumab	5 M	Ulcers (entire oral mucosa,	Epithelial necrosis,	Dsg1 (-), Dsg3 (-)	Erythromatous	PR	Oral prednisolone	Resolved after
					lip, tongue)	Lichenoid lymphocyte		papule			2 weeks
Enomo to	52/M	Lung adenocarcinoma	Nivolumab	5.5 M	Erosion (buccal, mouth floor,		Dsg1 (-), Dsg3 (-)	None		Oral prednisolone	Resolved within
[107]					gingiva)		BP180 (+)				3 weeks
Economopoulou	66/M	OSCC	Nivolumab	8 cycles	Ulcers (lower lip)					Bethamethasone cream	Responded well
[109]										ICI continued	
Shazib	82/F	Melanoma	Pembrolizumab	1 dose	Lichenoid (buccal, tongue)			Papular rash	PR	Topical DEX	[12/13 of patients
[108]								Knee arthralgia			reported greater than
	68/M	NSCLC	Pembrolizumab	9 doses	Lichenoid (buccal, tongue)	Lichenoid mucositis		None	PR	Topical DEX	80% improvement in pain scores,
	43/F	Melanoma	Pembrolizumab	2 doses	Lichenoid (palate, buccal,			Papular rash,	PR	Topical DEX	but there was
					tongue)			Adrenal crisis		Topical FLUOC	minimal
	57/M	Melanoma	Nivolumab	11 doses	Lichenoid (tongue)			Papular rash,	PR		objective
								Acute nephritis		Topical CLO	improvement]
	76/M	NSCLC	Nivolumab	6 doses	Lichenoid (tongue)			Papular rash,	PD	DEX	
								Diarrhea			
	70/M	OSCC	Pembrolizumab	4 doses	EM-like (tongue,			Dermatitis	PR	Topical DEX	
					buccak mucosa, lip					Prednisolone	
	73/F	NSCLC	Nivolumab	8 doses	Lichenoid (palate, buccal)	Lichenoid mucositis		Pneumonitis	PR	Topical CLO	
								Vaginal ulcers			
	57/M	Colon cancer	Pembrolizumab	1 dose	Acute GVHD			Papular rash	PD	Topical DEX	
					reactivation (palate,					M-prednisolone	
					tongue, buccal, lip)						
Phemphigoid with oral lesions											
Zumelzu	83/F	Melanoma	Pembrolizumab	16.5 M	MMP, erosion, blister	Subepithelial cleavage	DIF (+)	No skin lesions	CR	Doxycycline	Controled MMP
[63]					of gingiva	Perivascular infilitrate of	BP180 (-)				within 2 weeks
						lymphocytes and histiocytes	BP230 (-)				
Haug	62/M	Merkei cell carcinoma	Pembrolizumab	3.3 M	MMP, erosion, aphthous		DIF (+), IIF (+)	None		ICI discontinued	Erosion healed after
[125]					ulcers (tongue, buccal)		BP180 (+)			Doxycycline	6 weeks
										Topical mometasone	
Naidoo	80/M	Melanoma	Nivolumab	6 M	BP, bucal mucosa	Subepithelial vesicular	BP180 (+)		CR	Topical tacrolimus	
[121]						dermatitis with	BP230 (+), DIF (+)			and DEX	
						eosinophils					
	78/F	Melanoma	Durvalumab	13 M	BP, bucal mucosa	Subepithelial cleft	BP180 (+)		PR	Topical steroids	
							BP230 (+), DIF (+)				
Hwang	68/M	Melanoma	Pembrolizumab	19.5 M	BP, bucal mucosa	Dermal chronic	DIF (+)	Papules on trunks,	PR	Topical methyl-	Responded promptly
[122]						inflammation with		backs, legs		prednisolone	
						eosinophils				Doxycycline	
	72/M	Melanoma	Pembrolizumab	6.8 M	BP, bucal mucosa	Subepithelial blister	DIF (+)	Excorated blisters	PD	ICI stop, Doxcycline	
						with eosinophils		on trunk, back, legs		Topical prednisolone	
										Methotrexate	
Jour	63/M	HNSCC	Nivolumab	3.5 M	BP, bucal mucosa	Subepithelial blisters	DIF (+)	Vesiclees and bulae	PD	Topical fluocinonide	Progressive
[123]						with eosinophils		on neck, chin, trunk,		Oral prodnisolone	improvement
								four extremities		ICI discontinued	
Sowerby	80/M	Lung adenocarcinoma	Nivolumab	20 M	BP, gingival bulla	Subepithelial vesicle	DIF (+)	Vesicles and bullae	CR	ICI discontinued	Cleared within 2
[124]						with eosinophils	Dsg1 (+), BP180 (+)	on 4–5% of body		Oral steroids, Rituximab	months by rituximab
								surface			

Phemphigoid with oral lesions											
Wang	70/M	Melanoma	Pembrolizumab	35 cycles	BP, hard palate	Detachment of the	DIF (+)	Erosions (trunk, limbs)	CR	Oral prednisolone	Disappered 4 weeks later
[126]						epidermis		Blisters (hands, legs)			
Sadik	62/M	Melanoma	Pembrolizumab	6.8 M	BP, vesicular lesions of the oral mucosa	Interface dermatitis,	IIF (IgG+) BP180 (+),	Scattered skin papules	CR	Topical clobetasol, Oral prednisolone	Minor alleviation
[127]						Focal epidermal	BP230 (+)	with central vesicles		, Rituximab	
						necrosis					
	76/M	RCC	Nivolumab	4.8 M	BP, vesicular and white	Lichenoid interface	DIF (C3+), IIF (IgG+)	Palmoplanter hyper-	SD	Topical clobetasol (skin)	Alleviated but not completely
					reticular lesions of the	dermatitis	BP180 (+), BP230 (+)	keratosis, Polygonal		Topical triamcinolone,	resolved
					oral mucosa			papules, and vesicles		Dexpanthenol	
Paraneoplastic pemphigus											
Yatim	64/M	Cutaneous SCC	Pembrolizumab	0.7 M	Paraneoplastic pemphigus,	Suprabasal acantholysis	DIF (+)	Extensive cutaneous		ICI discontinued	Complete healing
[130]					blisters, pustules,	Intraepithelial blisters	Dsg1 (+), Dsg3 (+)	involvement		Oral prednisolone	
					Severe stomatitis						

ICIs, immune checkpoint inhibitors; irAEs, immune-related adverse events; M, month; CR, complete response; PR, partial response; SD, stable disease; PD, progressive disease; RCC, renal cell carcinoma; NR, not reported; DEX, Dexamethasone; Dsg1, desmoglein1; Dsg3, desmoglein 3; OSCC, oral squamous cell carcinoma; NSCLC, non-small cell lung cancer; FLUOC, flucinonide; CLO, Clobetasol; GVHD, graft-versus-host disease; DIF, direct immunofluorescence; IIF, indirect immunofluorescence; MMP, mucous membrane pemphidoid; BP, bullous pemphigoid; HNSCC, head and neck squamous cell carcinoma.

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
