# Peer review of "Oral Immune-Related Adverse Events Caused by Immune Checkpoint Inhibitors: Salivary Gland Dysfunction and Mucosal Diseases"

_cancers, 2022, doi:10.3390/cancers14030792_

Round 1

Reviewer 1 Report

Thank you for submitting your manuscript titled "Oral immune-related adverse events caused by immune checkpoint inhibitors: salivary gland dysfunction and mucosal diseases" to our journal. I thoroughly enjoyed reading the review paper and I find it remarkably comprehensive and well-written. It should receive the highest consideration for publication. I have but a few minor suggestions:

1) It is clear to me after reading the summary/abstract that one of the objectives, if not the main focus, is to describe the irAE of ICI use in HNSCC. Yet this is not apparent in the title. I would urge the authors to consider modifying it.

2) The focus of the review paper will dictate the appropriate audiences- if in fact the paper is geared towards clinicians (OTO surgeons, oncologists) then section 2 is appropriate in length and detail. Otherwise, most of the first several paragraphs describe fundamental knowledge that should be omitted to limit the length of the article. I would argue that only starting at line 134 that the translational component begins. The authors then go at length describing the experiences in other disease processes (melanoma, colorectal cancer, osteosarcoma in animal models).

3) has there been work done in mucosal melanoma

4) in my proof, line 280 is incomplete and lines 281-283 are completely missing

5) line 308 should read HNSCC not NSCC

Author Response

Reviewer 1  Comments and Suggestions for Authors

  • It is clear to me after reading the summary/abstract that one of the objectives, if not the main focus, is to describe the irAE of ICI use in HNSCC. Yet this is not apparent in the title. I would urge the authors to consider modifying it.

Response: Thank you for your comment. As indicated by the reviewer, one of the objectives, if not the main focus, is to describe the irAE of immune checkpoint inhibitor (ICI) use in HNSCC. However, the main purpose is to explain the ICI-induced irAEs that occur in the oral cavity by ICI treatment of various types of malignancies, including HNSCC, and is not necessarily limited to HNSCC. Indeed, many oral irAEs were reported in ICI-treated patients with malignancies other than HNSCC (Table 1, Figure 2, Table 2). In revised version, we have added a statement “ Treatment of various malignancies, including HNSCC, with ICIs may result in the appearance of oral irAEs” in abstract, introduction and conclusion (page 1, lines 14-15, 28-29; page 2, lines 78-79; page 19, lines 476-477). We would like to keep the current title.

  • The focus of the review paper will dictate the appropriate audiences- if in fact the paper is geared towards clinicians (OTO surgeons, oncologists) then section 2 is appropriate in length and detail. Otherwise, most of the first several paragraphs describe fundamental knowledge that should be omitted to limit the length of the article. I would argue that only starting at line 134 that the translational component begins. The authors then go at length describing the experiences in other disease processes (melanoma, colorectal cancer, osteosarcoma in animal models).

Response: The appropriate audiences are clinicians (OTO surgeons and oncologist), but also oral healthcare professionals who treat all types of oral diseases. This has been stated in introduction (page 2, lines 80-81).  ICI-induced irAEs occurred in various organs, but there must be a common mechanism for inducing such conditions. As shown in sections 4 and 5, ICI-induced irAEs may differ from known oral autoimmune diseases in histological and immunological findings, even with the same clinical characteristics. Therefore, in order to understand these differences, it is essential to first explain the possible mechanism by which ICI-induced oral irAEs occurs, as shown in other reviews. This section describes current knowledge of CTLA-4 and PD-1/PD-1 signaling pathways, broadly divided into T cell and B cell signaling. Section 2, including Figure 1, is an important part of this article.

  • has there been work done in mucosal melanoma

Response: Thank you for the comment. Mucosal melanoma should also be a target for ICIs. However, this time mucosal melanoma is not the main subject of this article and is not included in this article.

in my proof, line 280 is incomplete and lines 281-283 are completely missing

Response: Thank you for your comment. This has to be rectified. Line number is incorrect, but the sentence from 280 to 283 is not interrupted.

  • line 308 should read HNSCC not NSCC

Response: Thank you for your comment. This has been rectified (page 11, line 318).

Reviewer 2 Report

In the present review, authors have documented the immune adverse effects of immunotherapy treatment in HNSCC patients which may help in early diagnosis and efficient treatment. I have several reservations, my comments are appended as below:

  1. Sicca syndrome: please elaborate before using this term for the first time for general readers.
  2. Introduction- every statement should be justified with competent reference, bulk citing should be avoided.
  3. Provide examples of anti-PD-1, PD-L1, and CTLA-4 antibodies approved by the FDA (line 40).
  4. While discussing the side effects, authors should point out which type of immunotherapy treatment has a history of most adverse effects. In addition, the adverse effects are also noted in previous literature as secondary effects in PMID: 33076303. Authors should discuss in brief add descriptions.
  5. While referring to other cancer cases, the authors should exactly mention the subtype. Just mentioning ‘cancer’ would be vague. For instance, lines 134-135.
  6. Table 1 and 2: authors seem to add comprehensive details but the statistical inference (HR, P value) is missing. Authors should attempt to add the information.
  7. Line 266, reference 75- this is an interesting finding. The authors should elaborate on the cancer type.

8. Authors should also provide information on commonly used drugs to mitigate the toxic effects of immunotherapy, particularly for HNSCC.

  1. There should be a ‘future directions’ section.

Author Response

Reviewer 2   Comments and Suggestions for Authors

  1. Sicca syndrome: please elaborate before using this term for the first time for general readers.

Response: Thank for your valuable comments. We have added a sentence showing the correlation between primary Sjogren syndrome (pSS) and ICI-induced Sicca syndrome (page 11, line 336).

  1. Introduction- every statement should be justified with competent reference, bulk citing should be avoided.

Response: As indicated by the reviewer, every statement should be justified with component reference, but not bulk citing. However, some reviews are needed to summarize the past treatment process of HNSCC and to optimize the volume of articles. In the revised version, we have added several references related with the treatment of HNSCC (ref. 3 and 4).

  1. Provide examples of anti-PD-1, PD-L1, and CTLA-4 antibodies approved by the FDA (line 40).

Response: Following the suggestion, we have added text and references describing the FDA-approved ICIs (nivolumab and pembrolizumab) for HNSCC (page 2, line 44, ref. 1).

  1. While discussing the side effects, authors should point out which type of immunotherapy treatment has a history of most adverse effects. In addition, the adverse effects are also noted in previous literature as secondary effects in PMID: 33076303. Authors should discuss in brief add descriptions.

Response: Historically, it has been stated that anti-CTLA-4 induced irAEs more frequently than anti-PD-1 and anti-PD-1 antibodies (page 5, lines 1-2, ref. 43). In addition, the paper PMID: 33076303 was cited in the conclusion (ref. 131).

  1. While referring to other cancer cases, the authors should exactly mention the subtype. Just mentioning ‘cancer’ would be vague. For instance, lines 134-135.

Response: lines 134-135, “cancer” has changed to “melanoma” (page 3, line 142).

  1. Table 1 and 2: authors seem to add comprehensive details but the statistical inference (HR, P value) is missing. Authors should attempt to add the information.

Response: No comparison was made on these tables, so the HR and P-values were not determined.

  1. Line 266, reference 75- this is an interesting finding. The authors should elaborate on the cancer type.

Response: The cancer type (MCA 205 mouse sarcoma) in the animal experiment was added to the text (page 6, line 275).

  1. Authors should also provide information on commonly used drugs to mitigate the toxic effects of immunotherapy, particularly for HNSCC.

Response: Table 1 shows that irAEs, which are frequently seen in HNSCC patients treated with ICIs, are dermatological, endocrine, and gastrointestinal disorders. The treatment of these irAEs is not included in this article. Instead, it covers the treatments that are applied to irAEs in the oral cavity, and the details are shown in Table 2. The treatment begins with topical corticosteroids in most patients, if improvement is not observed, followed by systemic administration of corticosteroids and discontinuation of ICI.

  1. There should be a ‘future directions’ section.

Response: Instead of the section of future directions, some comments stating future directions have been added to the conclusion with reference (page 20, lines 485-489, ref. 131)

Reviewer 3 Report

This is an interesting review about oral immune-related adverse events caused by immune-checkpoint inhibitors, with a focus on salivary gland dysfunction and mucosal diseases.

The paper is well written. The authors described molecular mechanisms of immune-related adverse events.

I think that the authors should state in the title that the review mainly describe molecular pathways more than clinical parameters.

Author Response

Reviewer 3    Comments and Suggestions for Authors

This is an interesting review about oral immune-related adverse events caused by immune-checkpoint inhibitors, with a focus on salivary gland dysfunction and mucosal diseases.

The paper is well written. The authors described molecular mechanisms of immune-related adverse events.

I think that the authors should state in the title that the review mainly describe molecular pathways more than clinical parameters.

Response: Thank you for your positive comments. As indicated by the reviewer, one of the purposes of this article is to explain the possible molecular mechanism by which ICIs can induce oral irAEs. However, Sections 3, 4, and 5 consist mainly of clinical findings of affected patients. We would like to keep the current title to avoid lengthening the title with the addition of terms such as molecular mechanism and clinical study.

Reviewer 4 Report

Yura et al. Immune-related adverse events caused by immune checkpoint inhibitors. Salivary Gland Dysfunction and Mucosal Diseases

Yura et al. targeted a hot topic and lumped together a lot of information. They did a great work on all the immune-related adverse events in ICI therapy. But at all the review offers to many information. The should critically go through their text and focus on the HNSCC relevant information.

In detail they should follow the following points.  

1) All abbreviations (e.g. CTLA-4) have to be shown in a legend separately, esp. from Fug. 1.

2) Make formulations easier. As an example “The germinal center (GC) plays an important role in the proliferation and differentiation of B cells. B cell activation usually requires signals from B cell receptors (BCR), B cell co-receptors, and CD4+ helper T cells.” This goes through the entire text.

3) In Fig, 1 A LN the CD28 on CD8+ T cell has no counterpart as well as in A periphery PD-1 on Treg cell and in B GC CTLA-4 on Tfr cell.  Is that right?

4) B lymphocyte and cancer cells have PD-L1?

5) line 168-170: “….IgG1 antibody, the Fc portion of antibody in immunocomplexes can mediate antibody-dependent cellular cytotoxicity (ADCC), antibody dependent cellular phagocytosis (ADCP), and complement-dependent cytotoxicity (CDC)”.    That is Okay. But now, the following has to be written according to these three topics. Otherwise the reader is lost in terms and phrases.

6) line 177:  “no difference in the levels of Th17“. Do you mean in numbers of these lymphocytes?

7) line 183-185: “Similarly, treatment of colorectal cancer-bearing mice with anti-CTLA-4 antibodies decreased Foxp3+/CD4+ Treg cells, but increased CD4+ T and CD8+ T cells and the expression levels of pro-inflammatory Th1/M1-related cytokines IFN-g, IL-1a, IL-2, and IL-12 [48]. “ What is your point? Make a hierarchy as such first level the four situations you show in fig. 1  A LN, A periphery, B GC and B periphery, second level activating interactions, third inhibitory interactions and fourth autoimmunological/-aggressive outcome.

8) line 249-268: ” Bacterial species such as Bacteroides, Clostridium, and Faecalibacterium were shown to 249 expand Treg and produce anti-inflammatory cytokines. Recently, the relationship be- 250 tween the gut microbiota and irAEs has been investigated [72,73]. Dubin et al. [72] found 251 that species belonging to the Bacteroidetes phylum conferred resistance to colitis that de- 252 veloped after anti-CTLA-4 antibody therapy, suggesting that expansion of these bacteria 253 helps to prevent the development of ICI-induced colitis. The microbiota plays an im- 254 portant role in the endogenous synthesis of water-soluble B vitamins. A reduced capacity 255 for microbe-mediated production of B vitamins and polyamine transport may lower the 256 threshold for the onset of immune-mediated colitis. The group 3 innate lymphoid cells 257 (ILC3s) are the major innate lymphoid cell type involved in the immunopathology of ICI- 258 induced colitis. Lactobacillus reuteri (L.reuteri) administration markedly decreased the mu- 259 cosal numbers of ILC3s and therapeutically prevented colitis in the ICI-treated mice. This 260 suggests that L. reuteri affects the immunopathology of ICI-associated colitis primarily by 261 altering the local number of ILC3s. [74]. Oral administration of Bacteroides fragilis, Bac- 262 terodes thetaiotaomicron, and Burkholderia cepacia was found to restore the antitumor effects 263 of anti-CTLA-4 antibody in mice treated with a broad spectrum antibacterial cocktail. 264 These bacteria enhanced antitumor immunity by inducing the Th1 immune response in 265 regional lymph nodes and promoting DC maturation [75]. Some oral bacteria migrate to 266 the intestinal tract. Therefore, oral bacteria may be useful to alleviate the colitis caused by 267 ICIs [76].“ Reduce to a few sentences and focus on HNSCC.

9) Line 296-301   “Compared with HNSCC, ICI treatment for melanoma (CheckMate 037) was started 296 in advance, in which the overall frequency of nivolumab-induced irAEs was 59% (157/268) and that of grade 3-4 was 9% (22/268) (Checkmate 037 2015) [13]. In another study of 278 melanoma patients, pembrolizumab (Check Mate 006) induced irAEs in 221 (79.5%) pa tients and grade 3-5 occurred in 37 (13.3%) patients (Keynote 006 2015) [12]. In 418 lung cancer patients treated with nivolumab, (Checkmate 057), irAEs occurred in 283 (68%) patients and grade 3-4 occurred in 44 (10%) patients [81].” Focus on HNSCC, no melanoma or ling cancer!

10) line 316-324: Skip that lines and insert one sentence about Sjögren disease in the follwing.

11) Section 4 has to be cut down, To many details which could not be grasped by the reader.

12) As the conclusion focus on HNSCC  do so for the whole paper.

Author Response

Reviewer 4  Comments and Suggestions for Authors

Yura et al. targeted a hot topic and lumped together a lot of information. They did a great work on all the immune-related adverse events in ICI therapy. But at all the review offers to many information. The should critically go through their text and focus on the HNSCC relevant information.

Response: Thank you for valuable comments. As indicated by the reviewer, this article contains a lot of information. This is because the purpose is to explain the ICI-induced irAEs that occur in the oral cavity by ICI treatment of various types of malignant tumors including HNSCC. The appropriate audiences are clinicians (OTO surgeons and oncologist), but also oral healthcare professionals who treat all types of oral diseases. Therefore, the text is not limited to HNSCC, but focused on ICI-induced oral diseases. This statement has been added to the text (page 2, lines 80-81).

  • All abbreviations (e.g. CTLA-4) have to be shown in a legend separately, esp. from Fug. 1.

Response: Following the suggestion, we have changed the lines of the legend of the figure.

  • Make formulations easier. As an example “The germinal center (GC) plays an important role in the proliferation and differentiation of B cells. B cell activation usually requires signals from B cell receptors (BCR), B cell co-receptors, and CD4+ helper T cells.” This goes through the entire text.

Response: In order to explain the effect of anti-PD-1 antibody on the B cell pathway in the next subsection 2.2, it is necessary to briefly describe the known interaction between GC B cells and Tfh. In revised version, a related reference was added at the end of the sentence (ref. 33).

  • In Fig, 1 A LN the CD28 on CD8+ T cell has no counterpart as well as in A periphery PD-1 on Treg cell and in B GC CTLA-4 on Tfr cell.  Is that right?

Response: Ideally, it is better to provide some figures showing CD8+ T cells that are activated via CD28 and blocked via CTLA-4 during anti-CTLA- treatment. However, in a limited figure, as indicated by the reviewer, the DC counterpart to CD28 on CD8+T cells is not shown in figure 1A. Instead, inhibitory signals have been shown between CD8+ T cells and DC targeted by anti-CTLA-4 antibody. There is a limit to including everything in the figure, and instead, the text described the flow of tumor antigen presentation, recognition, and effector cell activation (page 3, lines 110-112, 116-119, 123-125). In “PD-1 on Treg cell and in B GC CTLA-4 on Tfr cell”, the actual role of PD-1 expressed on Treg has not been confirmed, but it may be blocked by anti-PD-1 antibodies. Tfr is a T follicular regulatory cell and expresses CTLA-4. This has been explained in text (page 3, lines 139-141).

  • B lymphocyte and cancer cells have PD-L1?

Response: Yes. This has been shown in previous literatures (ref. 24, 33).

  • line 168-170: “….IgG1 antibody, the Fc portion of antibody in immunocomplexes can mediate antibody-dependent cellular cytotoxicity (ADCC), antibody dependent cellular phagocytosis (ADCP), and complement-dependent cytotoxicity (CDC)”.    That is Okay. But now, the following has to be written according to these three topics. Otherwise the reader is lost in terms and phrases.

Response: Following the suggestion, the statement “These actions may contribute to IgG1-mediated cytotoxicity” has been added (page 5, lines 181-182)..

  • line 177:  “no difference in the levels of Th17“. Do you mean in numbers of these lymphocytes?

Response: According to a report by reference ref 44 (revised, ref. 48), PBMC or sorted CD4+ T cells obtained from patients treated with ICI were activated in vitro. Flow cytometry analysis of these cells revealed an increase in the proportion of CD3+CD4+IL-17+ cells.

  • line 183-185: “Similarly, treatment of colorectal cancer-bearing mice with anti-CTLA-4 antibodies decreased Foxp3+/CD4+ Treg cells, but increased CD4+ T and CD8+ T cells and the expression levels of pro-inflammatory Th1/M1-related cytokines IFN-g, IL-1a, IL-2, and IL-12 [48]. “ What is your point? Make a hierarchy as such first level the four situations you show in fig. 1  A LN, A periphery, B GC and B periphery, second level activating interactions, third inhibitory interactions and fourth autoimmunological/-aggressive outcome.

Response: Figure 1A shows the pathway of T cells, and Figure 1B shows the pathway of B cells in which T cells and B cells are activated and differentiate into effector cells. In reference 48 (revised ref 52), Fiegle et al. investigated the effect of anti-CTLA-4 antibody on colon cancer model. When applied to figure 1A, anti-CTLA-4 antibody blocks the inhibitory checkpoint protein CTLA-4 on Treg, activate DC in lymph nodes (LNs), activate CD8+ T cells and its antitumor capacity (in the periphery). It also activates CD4+ cells. These cells produce pro-inflammatory cytokines such as IFN-g, IL-1a, IL-2, and IL-12.

  • line 249-268: ” Bacterial species such as Bacteroides, Clostridium, and Faecalibacterium were shown to expand Treg and produce anti-inflammatory cytokines. Recently, the relationship between the gut microbiota and irAEs has been investigated [72,73]. Dubin et al. [72] found that species belonging to the Bacteroidetes phylum conferred resistance to colitis that developed after anti-CTLA-4 antibody therapy, suggesting that expansion of these bacteria helps to prevent the development of ICI-induced colitis. The microbiota plays an important role in the endogenous synthesis of water-soluble B vitamins. A reduced capacity for microbe-mediated production of B vitamins and polyamine transport may lower the threshold for the onset of immune-mediated colitis. The group 3 innate lymphoid cells (ILC3s) are the major innate lymphoid cell type involved in the immunopathology of ICI-induced colitis. Lactobacillus reuteri (L.reuteri) administration markedly decreased the mucosal numbers of ILC3s and therapeutically prevented colitis in the ICI-treated mice. This suggests that L. reuteri affects the immunopathology of ICI-associated colitis primarily by altering the local number of ILC3s. [74]. Oral administration of Bacteroides fragilis, Bacterodes thetaiotaomicron, and Burkholderia cepacia was found to restore the antitumor effects of anti-CTLA-4 antibody in mice treated with a broad spectrum antibacterial cocktail. These bacteria enhanced antitumor immunity by inducing the Th1 immune response in 265 regional lymph nodes and promoting DC maturation [75]. Some oral bacteria migrate to the intestinal tract. Therefore, oral bacteria may be useful to alleviate the colitis caused by ICIs [76].“ Reduce to a few sentences and focus on HNSCC.

Response: It is noteworthy that some studies have proved evidence for the connection of gut microbiome and ICPs-induced colitis, being a developing new field of ICI-induced irAEs (ref. 79). In the future, it may be associated with the onset of oral irAEs in the mechanism. Therefore, this part is important.

  • Line 296-301   “Compared with HNSCC, ICI treatment for melanoma (CheckMate 037) was started 296 in advance, in which the overall frequency of nivolumab-induced irAEs was 59% (157/268) and that of grade 3-4 was 9% (22/268) (Checkmate 037 2015) [13]. In another study of 278 melanoma patients, pembrolizumab (Check Mate 006) induced irAEs in 221 (79.5%) pa tients and grade 3-5 occurred in 37 (13.3%) patients (Keynote 006 2015) [12]. In 418 lung cancer patients treated with nivolumab, (Checkmate 057), irAEs occurred in 283 (68%) patients and grade 3-4 occurred in 44 (10%) patients [81].” Focus on HNSCC, no melanoma or ling cancer!

Response: It is important to determine the frequency of irAEs in HNSCC patients. However, it is unclear how different the findings in HNSCC are from those of other types of malignancies. Therefore, we selected two typical malignant tumors, melanoma and lung cancer, and determined the frequency of irAE by ICI, and showed that there was no particular difference between the tumors. It may be necessary to consider melanoma and lung cancer for comparison with HNSCC.

  • line 316-324: Skip that lines and insert one sentence about Sjögren disease in the follwing.

Response: This part is needed to explain the differences and similarities between primary Sjogren syndrome (pSS) and ICI-induced Sicca syndrome.

  • Section 4 has to be cut down, To many details which could not be grasped by the reader.

Response: As mentioned above, the appropriate audiences are clinicians (OTO surgeons and oncologist), but also oral helthcare professionals who treat all types of oral diseases. Therefore, this article contains the major diseases of the oral cavity due to ICI treatment. With Sicca syndrome, patients complain of severe dry mouth and difficulty eating, which is an important oral disorder. This section is especially needed for oral healthcare professionals.

  • As the conclusion focus on HNSCC do so for the whole paper.

Response: This has been described in the response to 11). In addition, it can be stated that many oral ICI-induced irAEs are reported in patients with malignant tumors other than HNSCC (Table 1, Figure 2, Table 2). In revised version, we have added a statement “Treatment of various malignancies, including HNSCC, with ICIs may result in the appearance of oral irAEs” in abstract, introduction and conclusion ( page 1, lines 14-15, 28-29; page 2, 78-79; page 19, lines 476-477).

Round 2

Reviewer 2 Report

All my concerns are now addressed. 

Reviewer 4 Report

Revision acceptable.